# Enhancer RNA (eRNA) in Human Diseases

**DOI:** 10.3390/ijms231911582

**Published:** 2022-09-30

**Authors:** Yunzhe Wang, Chenyang Zhang, Yuxiang Wang, Xiuping Liu, Zhao Zhang

**Affiliations:** 1MOE Key Laboratory of Metabolism and Molecular Medicine, Department of Biochemistry and Molecular Biology, School of Basic Medical Sciences, Fudan University, Shanghai 200032, China; 2Department of Pathology, School of Basic Medical Sciences, Fudan University, Shanghai 200032, China

**Keywords:** enhancer RNA, eRNA, cancer, neurodegenerative disorder, cardiovascular disease, metabolic disease, transcriptional regulation

## Abstract

Enhancer RNAs (eRNAs), a class of non-coding RNAs (ncRNAs) transcribed from enhancer regions, serve as a type of critical regulatory element in gene expression. There is increasing evidence demonstrating that the aberrant expression of eRNAs can be broadly detected in various human diseases. Some studies also revealed the potential clinical utility of eRNAs in these diseases. In this review, we summarized the recent studies regarding the pathological mechanisms of eRNAs as well as their potential utility across human diseases, including cancers, neurodegenerative disorders, cardiovascular diseases and metabolic diseases. It could help us to understand how eRNAs are engaged in the processes of diseases and to obtain better insight of eRNAs in diagnosis, prognosis or therapy. The studies we reviewed here indicate the enormous therapeutic potency of eRNAs across human diseases.

## 1. Introduction

The human genome has ~98% non-coding regions, which used to be considered “junk DNA” [1,2]. Using high-throughput sequencing methods, a huge number of RNA transcripts from non-coding regions have been identified and defined as non-coding RNA (ncRNA) [3]. More and more studies have characterized the functional role of ncRNAs in diverse biological processes, suggesting that investigating ncRNA may aid us in comprehensively understanding the human transcriptome.

Non-coding RNA species, including transfer RNAs (tRNAs), small nuclear RNAs (snRNAs) and small nucleolar RNAs (snoRNAs), play constitutive roles in biological processes [4]. For example, tRNAs, typically 70–90 nucleotides (nt) in length, are essential in translation processes, mainly through transferring charged amino acids to initiate or elongate peptides [5]; snRNAs, approximately 150 nt in length, are assembled with many associated proteins to form small nuclear ribonucleoproteins (snRNPs), which dominate pre-mRNA splicing [6]; and snoRNAs, which range in length from 60 to 300 nt, guide the chemical modifications (e.g., pseudo-uridylation and methylation) of different types of RNA, including ribosomal RNAs (rRNAs), snRNAs and mRNAs [7,8,9,10]. Apart from the constitutive ncRNAs, there are other groups of ncRNAs, such as microRNAs (miRNAs), circular RNAs (circRNAs) and long intergenic non-coding RNAs (lincRNAs), which play a regulatory role in gene expression [4]. There are emerging studies revealing the relevance of miRNAs in gene regulation, mostly through their interaction with the 3′ untranslated region (3′ UTR) of target mRNAs, which can trigger mRNA degradation or translational repression [11,12,13]. Circular RNAs can be detected in eukaryotic cells with cell- or tissue-specific expression patterns. They are named after the covalently closed continuous loops which make them much more stable than linear RNAs in cells [14,15,16]. Due to the miRNA response elements they have, circRNAs are able to compete for miRNA binding sites [17]. Therefore, they act as miRNA sponges to induce continuous function loss for miRNAs [17,18]. Long intergenic non-coding RNAs also play pivotal roles in regulatory processes [19,20]. At the transcriptional level, lincRNAs can target transcriptional modulators and DNA duplex [21,22,23]. For example, X-inactive specific transcript (Xist), the key effector in X-chromosome inactivation, is a typical lincRNA. Through transcriptional repression, Xist can inactivate one of the two X-chromosomes entirely in somatic cells of female mammals [24,25]. At the post-transcriptional level, lincRNAs are involved in mRNA splicing, miRNA-mediated suppression and cell signaling-related translation [26,27,28,29]. One lincRNA called Plasmacytoma Variant Translocation-1 (PVT-1) has various impacts on tumorigenesis by downregulating tumor suppressor genes through encoding miRNAs [30,31]. At the epigenetic level, lincRNAs can participate both in covalent modifications through direct interactions with histone- and DNA-modifying factors and in ATP-dependent remodeling processes [20,32,33]. Taking *HOTAIR* as an example, it promotes tumor cell invasion by altering histone (H3K27) methylation patterns and increasing cancer metastasis in a manner dependent on PRC2 [31,34]. Overall, regulatory ncRNAs are involved in miscellaneous biological processes, varying from cell proliferation to pluripotency, showing a big potential in understanding and diagnosing cancers, inflammation responses and other diseases [17,35].

Non-coding RNAs play critical roles in the gene regulatory network (GRN), which is a term for dynamic interactions contributing to multiple biological processes. In the GRN, ncRNAs, together with proteins, mediate the accessibility and activation of DNA to increase or decrease the expression levels of target genes [36,37]. The ncRNA-mediated alteration of the GRN is associated with different biological processes, such as cell growth, cell differentiation and signaling responses, and thus contributes to human disease development [36,38]. For example, the lncRNA H19 regulates the expression of a cluster of genes in the imprinted genes, including insulin-like growth factor 2 (*Igf2*), solute carrier family 38 member 4 (*Slc38a4*) and paternally expressed gene 1 (*Peg1*), suggesting that the H19-mediated GRN is crucial in regulating the embryo’s development and imprinting pathologies [39,40,41].

Recent research found that one group of ncRNAs, named enhancer RNAs (eRNAs), are transcribed from enhancer regions and highly associated with epigenetic modifications of active enhancers, such as high H3K27ac and H3K27me1 [42,43]. Active enhancers recruit coactivators and transcription factors (TFs) to produce eRNAs [44], which, together with TFs and RNA-binding proteins (RBPs), can mediate gene expression [45,46]. The binding strength of RNA polymerase II also determines the transcriptional levels of their eRNAs [27,47]. For example, P300 binding to enhancers is crucial for the transcription of eRNAs [48]. CBP/P300 inhibitors leads to H3K27Ac reduction at enhancers and thus suppresses eRNA expression and downregulates eRNA target genes [43]. Chromatin loop formation is also vital for adequate eRNA expression. Direct RNA–protein interactions between eRNAs and boundary proteins such as CTCF may help to block loop-extrusion factors such as cohesin [49]. In addition, some tissue- or disease-specific TFs also regulate eRNAs to mediate the GRN. For example, breast cancer cells specifically express TF ESR1, together with its cooperating TF GATA3, inducing global transcriptional network changes by mediating enhancer accessibility as well as eRNA expression in breast cancer cells to promote tumorigenesis [50,51]. In humans, eRNA expression is tightly correlated with the expression of target genes. eRNAs are the markers for active enhancers in the local enhancer-promoter loop and long-range chromatin interactions [49]. Additionally, eRNAs can also serve as scaffolds to stabilize transcriptional complexes by directly mediating target genes [52]. Due to the high-affinity binding sites of eRNA and the potential for cooperative spreading of binding partners in the nucleus, they also perform as the key components in liquid–liquid phase separation [42,53]. Taken together, regulators of eRNAs, through general or specific manners, contribute to the alternation of the GRN by mediating expression in eRNA and eRNA target genes [48,54,55,56,57,58]. 

Increasing studies have begun to investigate eRNAs in diverse human diseases, such as breast cancers, gastric cancer and pancreatic cancer. The functional roles and clinical liability of eRNA are still continuously being explored in many diseases, particularly in some rare diseases. Herein, we summarized the molecular mechanisms and functional consequences of eRNAs across cancer types, neurodegenerative disorders, cardiovascular diseases and metabolic diseases and discussed the potential clinical utility of eRNAs as biomarkers in a range of diseases. 

## 2. Role of eRNAs in Cancers

Cancers are associated with multiple aberrant biological processes, including sustained proliferation, immune escape and apoptosis prevention [59,60]. The aberrant expression of eRNA dysregulates cancer-related genes to promote cancer signaling pathways and results in abnormal cellular responses in tumorigenesis [45,61,62]. In this section, we summarize the contribution and mechanisms of eRNAs in different cancer types, as well as their potential clinical utility for cancer diagnosis and therapies.

### 2.1. The Contribution and Mechanisms of eRNAs in Cancer

According to recent studies, eRNAs are highly engaged in oncogene activation and aberrant signaling pathways, which contribute to tumorigenesis [45]. The eRNAs transcribed from the super-enhancer regions on 8q24 serve as a good example of how this works [30,63,64,65,66]. Using the chromosome conformation capture (3C) technique, researchers have found long-range physical interactions between eRNAs from 8q24 and MYC, whose dysregulation is a significant symbol of multiple cancer types [63,67,68,69,70,71]. For example, colon cancer-associated transcript 1 (CCAT1), one well-studied eRNA transcribed from the 8q24 locus, can be activated through long-range interaction with MYC in many cancers, such as colorectal cancer (CRC), prostate adenocarcinoma (PRAD), breast invasive carcinoma (BRCA) and esophageal carcinoma (ESCA) [63,72,73,74]. The evidence has shown that CCAT1 plays its role through different molecular mechanisms in the nucleus and cytoplasm [72]. In the nucleus, CCAT1 directly binds to Asp-Glu-Ala-Asp (DEAD) box helicase 5 (DDX5), which acts as a coactivator involved in *MYC* gene transcriptional activation and is also able to promote tumor growth [72,75,76,77]. In the cytoplasm, CCAT1, acting as a miRNA sponge, competes with MIR-28-5P for miRNA binding sites to promote prostate cancerous processes (Figure 1B) [72]. Furthermore, in human CRC cells, a long isoform of CCAT1 called CCAT1-L has been observed to upregulate *MYC* and thus promote tumorigenesis. CCAT1-L achieves such functions by interacting with CTCF and modulating chromatin conformation (Figure 1B) [78]. As for the contribution of eRNAs in aberrant signaling responses, take the variant rs72725854 in a prostate cancer-specific enhancer at the 8q24 locus as an example. The eRNA expression level is significantly affected by this variant and can thus regulate neighboring lncRNAs such as PVT1 [66]. Bioinformatics analysis in human CRC has shown that its expression level is statistically associated with most of the genes within the TGFβ/SMAD and Wnt/β-Catenin pathways (Figure 1B) [66,79]. These pathways are vital to tumorigenesis by contributing to proliferation, differentiation and anti-apoptosis [80,81]. Taken as a whole, the variants, modifications or abnormal activation of 8q24 regions can induce overexpression of its transcribed eRNAs, which then trigger the pathogenesis of multiple human cancers, including CRC, PRAD, BRCA, ESCA and hematological malignancies [65,66,69,70,71,82]. 

The interaction of eRNAs with Bromodomain and Extraterminal (BET) proteins also showed some insights into cancer development. BET, a protein family containing BRD2, BRD3, BRD4 and BRDT, is regarded as an epigenetic regulator of transcription in carcinogenesis-related biological processes, including cell growth, differentiation and inflammation [83]. In different tissues or pathways, their mechanisms are varied and complicated. BET proteins (e.g., BRD4) can maintain eRNA over-synthesis and the continuous activation of NF-κB as well as the subsequent aberrant inflammatory responses or cell proliferation [84,85,86,87,88]. Recent studies have investigated the direct interplay between BRD4 and certain eRNAs, including eRNAs of MMP9 and CCL2. The tandem bromodomains of BRD4 were found to facilitate the BRD4–eRNA interactions. Conversely, these eRNAs also promote the functions of BRD4 by increasing its binding towards acetylated histone peptides and modulating the occupancy of the BRD4 enhancer to keep the gene active [89]. 

Another example is FOXP4-AS1, whose upregulation has been identified in different cancers, including hepatocellular carcinoma (HCC), CRC, PRAD and ovarian serous cystadenocarcinoma (OV) [90,91,92,93]. The mechanisms of FOXP4-AS1 in these cancers have also been well studied. For example, in PRAD, the expression of FOXP4-AS1 can upregulate its target gene *FOXP4* by acting as a miRNA sponge and sequestering miR-3184-5p. Researchers further determined that the *FOXP4*-AS1/FOXP4 axis can promote PRAD tumorigenesis [92]. In HCC, FOXP4-AS1 can recruit enhancers of zeste homolog 2 (EZH2) on H3K27me to downregulate the expression of zinc finger CCCH-type containing 12D (ZC3H12D), thus facilitating HCC growth [91]. Such cancer-specific mechanisms indicate a detailed and comprehensive perspective in further research. 

Apart from the detailed mechanisms of eRNAs mentioned above, more eRNAs are being detected and studied in different cancer types. By inducing *cIAP2* expression, BIR3C eRNA can affect *H. pylori* infection, which is one of leading causes of gastric cancer [94]. An immune-related eRNA called WAKMAR2 can regulate several downstream target genes in BRCA [95]. In non-small lung cancer cells, the eRNA TBX5-AS1 has been found to regulate the tumor progression through the PI3K/AKT pathway [96,97]. KLK3e, involved in androgen receptor-driven looping, has been discovered to selectively trigger gene activation in PRAD [98]. An important eRNA called NET1e has been observed to have overexpression patterns in PRAD, stomach adenocarcinoma (STAD), liver hepatocellular carcinoma (LIHC), kidney renal clear cell carcinoma (KIRC) and kidney renal papillary cell carcinoma (KIRP) [99,100]. All these examples, even not covering every pattern in eRNA regulation, are fulfilling the understanding of eRNAs in carcinogenesis and cancer growth, which may also shed light on more cancer treatment methods.

### 2.2. The Potential Clinical Utility of eRNAs in Cancer

Increasing evidence has also shown the significant roles of eRNAs in clinical applications, including acting as biomarkers and therapeutic targets. First, eRNAs can serve as potential biomarkers in diagnosis. For example, in CRC cells, the expression of the eRNA CCAT1 is correlated with differential grades, non-mucinous histology and tumor stages (III and IV) (Figure 1A) [101]. Researchers have also found analogous patterns in other cancer types such as BRCA, in which a correlation has been identified of high CCAT1 expression with lymph node metastases as well as all the diagnostic aspects mentioned above in CRC (Figure 1A) [73]. It should be noted that CCAT1 may provide a limited contribution in detecting CRC at an early stage [102]. In a more recent study, researchers discovered the association between the overexpression of *MYC* and CCAT1 upregulation before prostate tumor formation, signifying a novel tool for early PC identification [72]. This evidence suggests that eRNAs could be employed as biomarkers in cancer diagnosis. In addition, more studies have focused on the possible utility of eRNAs in cancer prognosis. CCAT1 has also shown its potential to be an independent indicator in CRC and PC to predict poor survival both in recurrence-free survival (RFS) and overall survival (OS) [72,101]. In addition, high expression of FOXP4-AS1 has also been reported to contribute to worse survival and a higher recurrence rate in HCC and CRC patients [91,93]. Bioinformatics methods applied to analyze the TCGA database have also revealed some promising biomarkers for further experiments. For instance, a pan-cancer analysis has displayed that the expression of eRNA SPRY4-AS1 in surgical specimens is associated with survivability in patients with HCC, glioblastoma multiforme (GBM), adrenocortical carcinoma (ACC), brain lower grade glioma (LGG) or mesothelioma (MESO) [103]. Such analysis of the association between eRNA expression levels and survival or relapse rate suggests that eRNAs can be good prognostic biomarkers.

The inhibition of oncogenic eRNA has potential utility in cancer therapy. Studies have found that FOXP4-AS1 knockdown by its targeted shRNA can inhibit CRC cell proliferation and induce apoptosis both in vitro and in vivo [93]. A similar study of eRNAs in BRCA revealed substantial and strong cell growth inhibition in the presence of NET1e knockdown [100]. Apart from eRNA depletion, the analysis of eRNA expression level and drug sensitivity also displayed therapeutic usage. For example, from a TCGA cohort, researchers have observed a strong connection between the response of anti-cancer drugs and eRNA expression through within pathways or cross-pathways, which indicates that changes in eRNA expression may work as a compliment in anti-cancer drug therapies [100]. Furthermore, it has been found that even more hopeful therapeutic potentials are related to BET proteins. In recent years, several BET inhibitors have been developed and tested in clinical trials [104]. These inhibitors all aim in targeting the BRD-acetyl binding pocket to prevent BRD4 from binding to enhancers or super-enhancers [85,104]. According to the discoveries in the direct interaction between certain eRNAs (i.e., MMP9 and CCL2) and BRD4, eRNAs could also be potent therapeutic targets for repressing carcinogenesis [89]. The novel clinical tool targeting eRNAs in this process, together with the inhibitors that have already been put into clinical trials, may shed light on the effect of eRNAs in cancer treatment. 

Although researchers have identified the potential clinical utility of eRNAs as biomarkers and therapeutic targets across different cancer types, more comprehensive trials are required [105]. As discoveries of eRNAs are centered in cancer-related studies, the significance of eRNAs in carcinogenesis and their cancer-related clinical relevance would be understood.

## 3. Role of eRNAs in Neurodegenerative Disorders

Neurodegenerative disorders are caused by the progressive loss of select vulnerable populations of neurons and result in a debilitating loss of sensory, motor and cognitive functions [106,107]. The disorders also share fundamental biological processes related to progressive neuronal dysfunction and death, abnormalities of autophagy and neuroinflammation [106]. Aberrant expression genes involved in those biological processes are vital for interested researchers. Being a critical regulatory element in gene regulation, the eRNAs that respond to neuronal depolarization are worth studying. Among the various neurodegenerative disorders, Alzheimer’s disease (AD), Parkinson’s disease (PD) and Huntington’s disease (HD) have been recently linked to eRNA dysregulation, which may provide a better knowledge of the disease processes and novel possibilities of utilizing eRNAs in clinical usage [108,109,110]. 

### 3.1. The Contribution and Mechanisms of eRNAs in Neurodegenerative Disorders

The role of eRNAs in neurodegenerative disorders has recently been observed through various bioinformatic and experiential methods. Several eRNAs are involved in neurodegenerative pathogenesis or regulating the expression of certain genes that have vital roles in these diseases. In addition, bioinformatics investigations have revealed broader expression patterns or enrichment of eRNAs in neurodegenerative disorders, extending the bounds of regulatory elements in neurodegenerative disorders. 

It is known that brain-derived neurotrophic factor (BDNF) signaling is impaired in AD brains, while increasing BDNF levels can improve learning and memory [111]. In the intergenic region interacting with BDNF, two putative eRNAs—designated Bdnf-Enh^g1^ and Bdnf-Enh^g2^—have been found to regulate BDNF expression by their transcribed eRNAs. More specifically, the expression levels of these putative eRNAs have been observed to have a dramatic increase in tandem with BDNF expression, especially in post-mitotic neuron cells. Further experiments have validated their role in mediating BDNF expression that contributes to dendritic growth [112]. The onset of neurodegenerative diseases has been linked to abnormal BDNF expression regulated by Bdnf-Enh^g1^ and Bdnf-Enh^g2^ [111,112,113]. Another study has shown the intermediary but important involvement of eRNAs in pathogenesis by observing the subsequent impact of DNA sequence variants in the enhancer region ANNCR. This change is associated with Apolipoprotein E (APOE), whose ε4 allelic form is a major risk factor for AD. AANCR is folded into an m^6^A R-loop and partially transcribed into a certain eRNA, which then effectively silences APOE expression and changes the susceptibility to AD [114]. An expression profile study has also revealed a significant enrichment of differentially expressed eRNAs at AD-associated enhancer regions [115].

Some studies have observed the eRNA involvement in other neurodegenerative diseases. For example, in dopamine neurons, several PD-associated variants on chromosome 17q21 have been linked to a putative eRNA expressed from intron 2 of the KANSL1 gene [116]. As KANSL1 dysregulation has been proved to disturb autophagy and thus induce memory impairment and neurodegeneration, the regulatory mechanism of this putative eRNA is worth investigating in PD-associated research [116,117]. Evidence in HD-related studies also displays the importance of eRNAs. For instance, researchers have found that decreased H3K27ac activity at super-enhancers in the R6/11 striatum is able to alter the eRNA production in HD-affected neurons [109]. RNA pol II is then reduced along HD downregulated genes, especially the voltage-gated potassium channels (e.g., Kcnab1, Kcna4 and Kcnj4), causing transcriptional dysregulation of relative neuronal genes and presumably increasing striatal vulnerability in HD [109,118]. 

Other neurological pathways or circuities—capable of affecting both neurodevelopment and neurodegeneration—have been demonstrated to be associated with eRNAs. For example, a brain-specific ultraconservative eRNA called Evf2 can regulate the expression of the homeodomain TF Dlx5/6 by recruiting DLX and MECP2 in their intergenic enhancer regions. Evf2 mutants reduce the GABAergic interneuron numbers in both early postnatal hippocampal and dentate gyrus and decrease synaptic inhibition in adults’ brains [119,120]. Causing an imbalance between excitatory and inhibitory signaling, GABA-regulated circuit dysfunction is well known for its versatile contribution to neurological diseases, which includes neurodevelopmental disorders such autism and Tourette’s syndrome, neurodegenerative disorders such as HD and epilepsy and other disorders such as dystonia and hepatic encephalopathy [119,121,122,123,124]. Further research into the mechanisms of eRNAs in neurodegenerative disorders is needed to provide a better understanding of these diseases on a comprehensive and more nuanced level.

### 3.2. The Potential Clinical Utility of eRNAs in Neurodegenerative Disorders

Some studies have further investigated the possible clinical usage of eRNAs in neurogenerative disorders. For example, eRNA impacts HD-related neuronal genes in the striatum, as exemplified in Section 3.1., implying that targeting striatal enhancers can improve eRNA expression levels, which is likely to prevent the repression of neuronal genes in HD patients [118]. In PD rat models, the eRNAs transcribed from super-enhancers (SEs) have also been examined using bioinformatics techniques. A cluster of PD-specific SEs could upregulate SNX5 to prompt ferroptosis levels through the endosomal sorting pathway in PD, which may serve as potential diagnostic markers and therapeutic targets for PD [125].

A more detailed example may shed light on future research directions. An eRNA transcribed from the enhancer region located 5.8–7.0 kb upstream of the mouse *neurogenin1* (*Neurog1* or *Ngn1*) gene, designated utNgn1, has been demonstrated to be necessary for effective *Ngn1* transcription. Additionally, researchers have identified the elements that can control utNgn1 transcripts and hence indirectly regulate *Ngn1* expression: Wnt signaling is capable of upregulating utNgn1 expression, while PcG proteins are capable of downregulating it [126]. As for the function of the *Ngn1* gene, the loss of its expression can generate the loss of deep-layer neurons according to *Ngn*-independent mechanism research [127]. *Ngn1* overexpression, on the contrary, promotes premature differentiation and has been considered a means of helping embryonic stem cells (ESCs) in their differentiation into induced neurons [107,126]. The potential therapeutic utility in such a scenario is to replace degenerative neurons with these induced neurons from ESCs [107]. A computational approach has demonstrated the possible contribution of eRNAs derived from super-enhancers in ESC differentiation [128]. Combining the results of these studies, regulating the eRNA utNgn1 to prompt the directional differentiation of ESCs clearly turns out to be potentially useful in stem cell therapy for neurodegenerative disorders. 

Thus far, the clinical utility of eRNAs, e.g., prognosis or target therapy, in neurodegenerative disorders is still largely unexplored. These studies have not only revealed the mechanisms of eRNAs in different neurodegenerative disorders but also provided potential approaches to utilizing eRNAs in these diseases. 

## 4. Role of eRNAs in Cardiovascular Diseases

Cardiovascular diseases (CVDs) occur in the blood circulatory system, including the heart and its associated blood vessels [129]. Some studies have revealed that TF genes such as *NKX2-5*, *MESP1*, *TBX5*, *MEF2c*, *HAND2*, *GATA4* and *SRF* play critical roles in cardiac gene regulatory networks (GRNs) [130,131]. Most importantly, the eRNAs involved in these cardiac GRNs can perturb them and modulate CVD development [130]. Meanwhile, eRNAs have also shown their potential clinical utility for CVD diagnosis and treatment [132].

### 4.1. The Contribution and Mechanisms of eRNAs in Cardiovascular Diseases

Transcriptional alteration can be broadly detected in cardiovascular morphogenesis. Recent studies have revealed that various genes are regulated by eRNAs in these cardiac processes [131]. An eRNA known as CARMEN is a typical example that sparked a flurry of investigation. It maps to the locus proximal to an important region that harbors a cardiac SRF/NKX2.5-bound enhancer in human cardiac precursor cells (CPCs). Computational calculation and experimental validation have corroborated CARMEN as a crucial upregulated eRNA that contributes to both CPC differentiation and cardiovascular pathologies in human hearts. The activity of CARMEN in CPC differentiation is in its interaction with SUZ12 and EZH2, components of polycomb repressive complex 2 (PRC2) [133,134]. In addition, one of CARMEN’s isoforms, named CARMEN3, has been observed to show significant upregulation in the pathologies of two different human CVDs, namely idiopathic dilated cardiomyopathy (DCM) and aortic stenosis (AOS) [134]. 

Enhancer RNAs are also associated with calcium-handling physiology crucial for maintaining a healthy cardiac rhythm and intact heart activities. In human hearts, the eRNA *RACER* improves the performance of T-box transcription factor 5 (*TBX5*), whose malfunction is responsible for cardiac conduction system (CCS) disease through GRN dysregulation. As an upstream regulator of *TBX5*, *RACER* maintains normal calcium kinetics by regulating the expression of the TBX5-dependent gene *Ryr2* by recruiting Pol2 and stabilizing the structure of the Pol2-*Ryr2* promoter [135,136]. 

Notably, the eRNAs transcribed from different strands of the same enhancer may oppositely exert their regulatory functions according to novel research on Intergenic Regulatory Element *Nkx2-5* Enhancers (IRENEs) [137]. These two eRNAs are both transcribed from the enhancer of *Nkx2-5*, whose mutation has long been viewed as the main cause of congenital heart defects (CHDs) [138,139]. One IRENE eRNA is on the same strand (SS), while the other is in the divergent direction (div). IRENE-SS functions as a typical eRNA to promote *Nkx2-5* transcription by enlisting NKX2-5 to its own enhancer. On the contrary, IRENE-div silences the *Nkx2-5* enhancer by recruiting the histone deacetylase sirtuin 1 (SIRT1) (Figure 2) [137]. The subcellular distribution of IRENEs exhibits specific patterns: the localization of IRENE-div is dependent on the stage of the transcript’s biogenesis, with the mature form previously in the cytoplasm and the immature form having a subcellular localization similar to that of IRENE-SS. In contrast, 60% of IRENE-SS was found in the nucleus, with 45% of it bound to chromatin [137].

### 4.2. The Potential Clinical Utility of eRNAs in Cardiovascular Diseases

Understanding the essential roles of eRNAs in CVDs will also pave the road for manipulating eRNAs in related clinical studies. To be specific, eRNAs may be used as therapeutic targets in corresponding cardiovascular pathologies and may be utilized to control the differentiation of precursor stem cells into wanted cardiac cells in regenerative medicine due to their critical roles in cardiac cell development.

For example, Wisper (Wisp2 super-enhancer–associated RNA) has been identified as a cardiac fibroblast (CF)-enriched eRNA that regulates the process of cardiac fibrosis [140]. CFs’ differentiation into myofibroblasts triggered by CF proliferation at first can initiate the pathological process of heart failure, resulting in diseases such as AOS and dilated cardiomyocytes (CM). Additionally, their pathologies are under apoptotic resistance and profibrotic signaling molecular influence [141]. Therapeutic depletion of Wisper in vivo has been proven to generate a significant decrease in the proliferation of CFs and can consequently inhibit cardiac fibrosis. The sequence of Wisper being relatively conserved in humans suggests that it is desirable to explore the potential utility of Wisper as an antifibrotic therapeutic target [140].

Similar clinical utility of eRNAs has also been demonstrated in stress-induced disease development. An eRNA named HERNA1 (hypoxia-inducible enhancer RNA 1) has displayed a great capacity to regulate its neighboring coding genes, such as *synaptotagmin XVII* and *SMG1*, by conferring hypoxia responsiveness towards them. Furthermore, as HERNA1 production is robustly induced by pathological stress, antisense oligonucleotides targeting HERNA1 can protect cells against stress-induced pathological hypertrophy and increase overall survival rate [142]. The curative effect of HERNA1 inhibition suggests that it could be a useful post-disease therapeutic target in pressure-overload CVDs, including aortic stenosis (AS) and hypertrophic cardiomyopathy (HCM) [142,143].

Aside from serving as therapeutic targets, eRNAs can also be applied in cardiac regenerative strategies to replenish lost cardiomyocytes [132]. Bvht, transcribed from an important locus enriched with heart-specific enhancers, has been demonstrated to be a requisite eRNA during ESC progression of nascent mesoderm toward a cardiac fate. This decisive step in ESC differentiation occurs by regulating Mesp1, a vital transcriptional factor marking the early cardiac precursor cells [132,144]. Meteor is another eRNA that performs analogous functions in ESC differentiation. Genetically or epigenetically manipulating Meteor has revealed its essential role in mesendoderm specification and subsequent cardiogenic differentiation [145]. Based on these findings, eRNA manipulation shows great promise for guiding the differentiation of precursor cells into desired cardiac cells, which could have intriguing applications in cardiac cell regeneration, providing more chances for cardiac cell therapies. 

## 5. Role of eRNAs in Metabolic Diseases

Dysfunction of major metabolic tissues, e.g., liver and pancreas, could result in a series of metabolic anomalies, which generate metabolic diseases such as obesity, type 2 diabetes (T2D) and hepatic steatosis [146,147,148]. These diseases also show a tight correlation with each other because of the whole integrity of the metabolic network, in which key regulatory elements actively interact [148]. As evidence for the critical involvement of regulatory elements in metabolic networks and related disorders is becoming increasingly abundant, eRNAs—a subset of these elements—have also come to be recognized as a subject worth investigating in metabolic diseases.

### 5.1. The Contribution and Mechanisms of eRNAs in Metabolic diseases

In studies investigating obesity genesis, researchers have found that an eRNA named Lnc-leptin, transcribed from the enhancer region upstream of the *leptin* (*Lep*) gene, is required for *Lep* expression. Lnc-leptin‘s regulatory effects are based on two different mechanisms: direct interaction with the *Lep* promoter and acting as a bridge between TFs and histone-modifying proteins [149]. Downregulated *Lep* expression, caused by Lnc-leptin deficiency, may incur a decrease in leptin, which controls appetite through its effect on the hypothalamus [150,151]. Maternal obesity-associated metabolic and epigenetic alterations can also lead to dampened responses in the offspring’s immune system by reducing the expression of IL6 eRNA and IL10RB eRNA in monocytes [152]. This evidence has emphasized the important engagement of eRNAs in metabolic mechanisms and obesity-associated immune responses. 

In lipid metabolism research on hepatic steatosis and non-alcoholic fatty liver disease (NAFLD), the liver expression of an eRNA designated OLMALINC (oligodendrocyte maturation-associated long intergenic noncoding RNA, located at human chromosome 10q24.31) has been proven to be tightly associated with statin use and serum triglycerides (TGs). Such connection relies on OLMALINC’s ability to promote the expression of its adjacent TG-gene recognized as stearoyl-CoA desaturase (SCD), which is known to be up-regulated in hepatic steatosis and NAFLD [153,154]. Notably, OLMALINC has a stable and spliced transcript with a poly-A tail, which indicates its possible secondary functions and a broad usage in future studies [153]. 

The regulatory function of eRNAs in some specific metabolism-related genes related to a broader range of metabolic diseases may provide a wider understanding of the involvement and even modification of eRNAs. For example, m^5^C-modified eRNAs in hepatic cells have been found to act as an intermediate substance in the peroxisome proliferator-activated receptor γ coactivator 1 alpha (PGC-1α)-related metabolic network. The dynamic change in the methylation and demethylation of PGC-1α, which plays vital roles in adapted metabolic responses, is caused by methyltransferase SET7/9 and Lysine-specific demethylase 1 (LSD1), corresponding with m^5^C eRNAs. Moreover, the eRNAs transcribed from the enhancer regions of PGC-1α’s target genes, such as 6-phosphofructokinase (*Pfkl*) and Sirtuin5 (*Sirt5*), have been proven to be inducible elements affecting carbamoyl-phosphate synthetase 1 (CPS1) activity and have thus been identified as important sensors of the metabolic state [155]. According to current knowledge on PGC-1α, it has been clearly linked to various metabolic complications, including obesity, T2D and hepatic steatosis [156]. Therefore, understanding how eRNAs work in PGC-1α pathways could aid the intervention of these disorders [155].

Studies have also explored the dysfunction of enhancers in rare diseases. For example, lipodystrophies are referred to as lipid-partitioning disorders. Their primary defect is the loss of functional adipocytes, resulting in ectopic steatosis, severe dyslipidemia and insulin resistance [157,158]. The activation of enhancers regulating the genes involved in lipid metabolic pathways can be influenced by *Tmem120a*, a transcription factor whose deficiency may suppress the expression of genes associated with lipodystrophies [159]. This evidence indicates the functional and clinical potential of eRNAs in rare diseases.

### 5.2. The Potential Clinical Utility of eRNAs in Metabolic diseases

Investigations of the eRNAs in metabolic diseases, particularly T2D, show that they are involved in various metabolic pathways. In adipocytes, a well-known nuclear receptor named peroxisome proliferator-activated receptor (PPARγ) regulates adipocyte biology by directly binding to its target genes [160]. An anti-diabetes drug designated rosiglitazone (rosi) can act as a high-affinity activating ligand for PPARγ to treat T2D [161,162]. By recruiting coactivators such as MED1, p300, and CBP, rosi can upregulate certain eRNAs and thus generate an enrichment of these eRNAs at PPARγ binding sites. For example, a series of bidirectional eRNAs transcribed from the enhancers upstream of the Fabp4 locus were observed to be upregulated by rosi. Additionally, 85% of these upregulated eRNAs have PPARγ bound nearby, although there are also rosi-induced downregulated eRNAs whose locations harbor binding sites for other TFs. A better understanding of the complex transcriptional changes caused by rosi may provide solutions and interventions for its severe side effects and toxicities [162]. 

Insulin responses are always impaired in T2D and subsequent obesity [163]. LncASIR, an eRNA also annotated as a four-exon transcript in the RefSeq database, is transcribed from a super-enhancer region upstream of *Lep* and has been demonstrated to be a required element for normal insulin signaling pathways. By using dcas9-KRAS and guide RNA, an LncASIR silencing experiment showed a considerable reduction in the insulin-induced gene program in adipocytes. Overexpression of LncASIR alone is not sufficient to increase insulin responses in T2D patients. If LncASIR is induced with its binding protein—polycomb repressive complex (PRC)—together, it might improve the clinical repair of insulin responses [164].

## 6. Conclusions and Perspectives

Enhancer RNAs are regulated by upstream factors and mediate the expression of their target genes in different diseases. They often harbor similar biological functions and molecular mechanisms. Comprehensive investigations of eRNAs could shed light on their roles in complex GRNs in human diseases. In addition, some eRNAs have shown particular and distinct mechanisms that may indicate new knowledge of their functions. Taking IRENEs as examples, the eRNAs transcribed from the same locus in different strands, designated IRENE-SS and IRENE-div, exert almost opposing functions in regulating their target gene, *Nkx2-5* (Figure 2) [137]. *Nkx2-5* mutation has long been viewed as the main cause of CHDs, and utilizing the counterbalance effect of IRENEs may be a breaking point for CHD treatment [139]. Thus far, the role of eRNAs in human diseases is still largely unexplored. Furthermore, some eRNAs have shown cascade amplifications in complicated GRNs. For example, the tandem bromodomains of BRD4 can facilitate its binding to certain eRNAs such as MMP9 eRNA. After the binding relation is formed, MMP9 eRNA starts to increase the binding ability towards acetylated histone peptides and modulates the occupancy of the BRD4 enhancer, which then maintains downstream genes’ activation [89]. As BRD4 is highly involved in the biological processes associated with carcinogenesis, such as cell proliferation, differentiation and inflammation, the serial amplification of BRD4-eRNA could provide novel insights about cancer development [86,87]. Similar to the mutual promotion of BRD4 eRNA binding, decreased H3K27ac at super-enhancers in the R6/11 striatum reduces the eRNA production in HD-affected neurons. Conversely, the downregulated eRNA production results in the transcriptional dysregulation of related neuronal genes and presumably increases striatal vulnerability in HD [109,118]. This draws attention to eRNAs’ involvement in multiple biological processes and demonstrates their critical roles in different human diseases. 

The systematic study of the functions of eRNA could promote eRNA in potential clinical applications (Figure 3, Table 1). The usage of eRNAs in cancer-related studies can be divided into three potential applications: diagnostic biomarkers, prognostic biomarkers and therapeutic targets. Thus far, these potential applications of eRNAs are still far away from clinical trials, let alone practical clinical uses. Before being applied in clinical trials, their medical potentials mentioned above still need more evidence based on comprehensively investigating their potential toxicity and limitations. Furthermore, eRNAs have been associated with clinical relevance, such as high expression of SPRY4-AS1 associated with poorer survival rates in cancers. The mechanisms and utility of these clinical associations also require further exploration in order to realize their potential application, which also calls for more research focusing on the uncharted capacity of eRNAs in their biological function exploration.

## Figures and Tables

**Figure 1 ijms-23-11582-f001:**
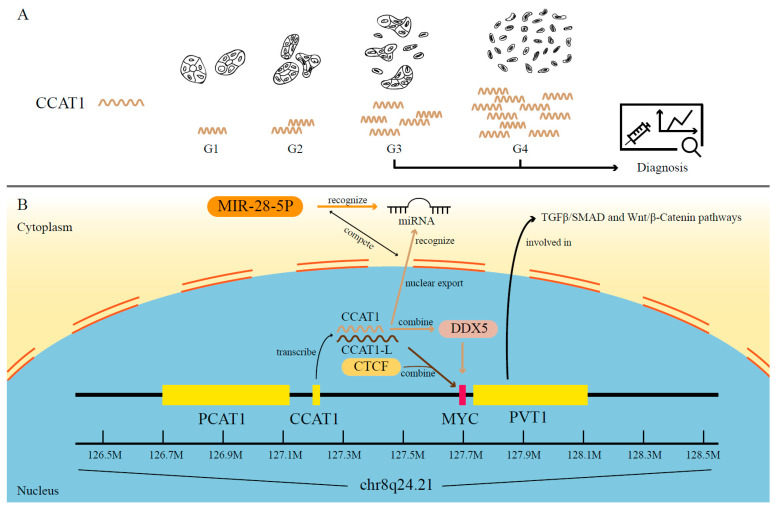
The mechanisms of eRNAs transcribed from chr8q24.21. (**A**) Utilizing CCAT1 as a diagnostic biomarker to distinguish CRC stages (III and IV). (**B**) The different mechanisms of CCAT1 and CCAT1L in the nucleus and the cytoplasm, as well as the impact on eRNA and follow-up pathways by a variant inside the PVT1 region.

**Figure 2 ijms-23-11582-f002:**
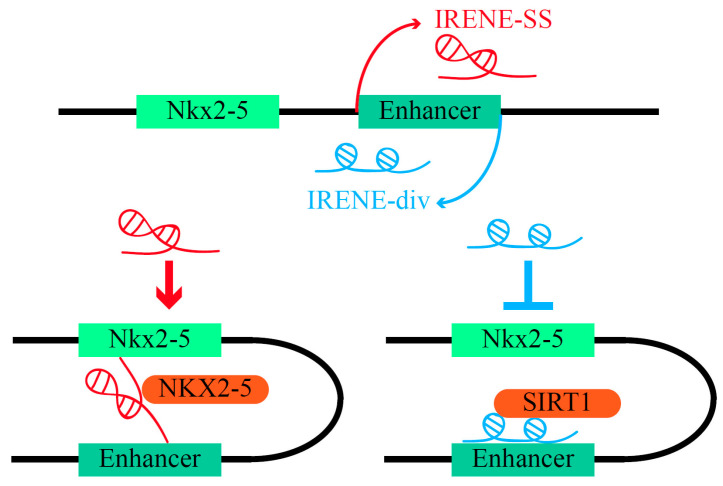
The opposing functions and mechanisms of IRENEs (IRENE-SS and IRENE-div) in regulating *Nkx2-5*. IRENE-SS, acting as a typical eRNA, enlists NKX2-5 and upregulates *Nkx2-5* expression. IRENE-div, on the contrary, recruits SIRT1 to *Nkx2-5* enhancer and silences *Nkx2-5* expression.

**Figure 3 ijms-23-11582-f003:**
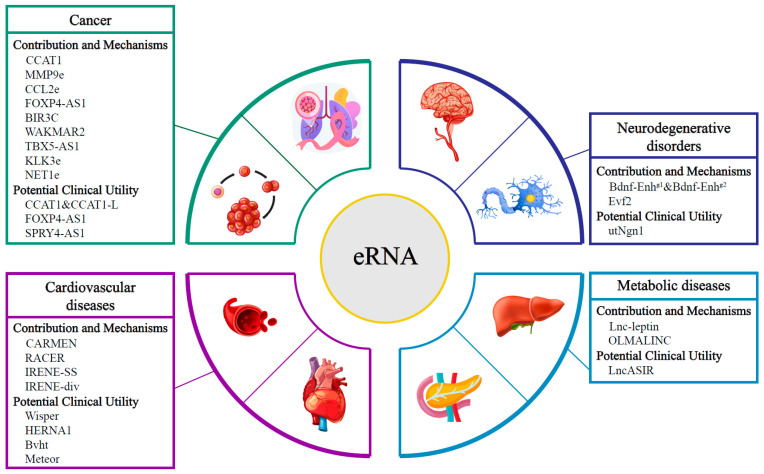
Identified eRNAs in cancer, neurodegenerative disorders, cardiovascular disease and metabolic diseases. In cancers, various eRNAs contribute to carcinogenesis, including CCAT1, MMP9e and other eRNAs mentioned in the figure. CCAT1 and FOXP4-AS1 also show their potential utility in cancer diagnosis, prognosis and therapy. In neurodegenerative disorders, Bdnf-Enh^g1^, Bdnf-Enh^g2^ and Evf2 are involved in the pathologies of AD, PD and/or HD. The eRNA utNgn1 has shown its possible medical usage. In cardiovascular diseases, CARMEN, RACER and IRENEs have demonstrated their contribution to different cardiac-associated pathways; Wisper, HERNA1, Bvht and Meteor may be applied in future therapies. In metabolic diseases, Lnc-leptin and OLMALINC can regulate gene expressions related to metabolic pathologies. Overexpression of LncASIR, together with inducing its binding protein, may be applied in T2D treatment.

**Table 1 ijms-23-11582-t001:** Detectable eRNAs, targets and experimental or clinical resources in different human diseases.

Identified eRNA	Disease Type	Target Gene	Experimental or Clinical Resources	Reference
CCAT1	CRC, PRAD, BRCA and ESCA	*MYC*	Patient and/or surgical samples and cell lines	[72,73,74,78]
MMP9 and CCL2 eRNA	BRCA, ESCA and PAAD	*Brd4*	Cell lines (SW480)	[86]
FOXP4-AS1	HCC, CRC, PRAD and OV	*FOXP4*	Patient and/or surgical samples and cell lines (PC-3, DU145, VCaP, LNCaP, RWPE-1, MHCC-97H, HepG2, LM3, SMMC-7721, DLD-1, HT-29, HCT116, SW480, Lovo)	[91,92,93]
BIR3C	Gastric cancer	*cIAP2*	Cell lines (AGS, MKN28, MKN45)	[94]
WAKMAR2	Breast cancer	*IL27RA*, *RAC2*, *FABP7*, *IGLV1-51*, *IGHA1* and *IGHD*	Patient samples and cell lines (MB-231, MCF7)	[95]
TBX5-AS1	Lung cancer non-small cells	*PI3K* and *AKT*	Patient samples and cell lines (16HBE, A549, H1299, NCI-H520)	[96,97]
KLK3e	PRAD	*PSA*	Cell lines (LNCaP, VCaP, COS-7)	[98]
NET1e	PRAD, STAD, LIHC, KIRC and KIRP	Downstream miRNAs (let-7e, miR-34, miR-98, miR-107, etc.)	Patient samples	[100]
Bdnf-Enh^g1^ and Bdnf-Enh^g2^	AD	*Bdnf*	Mice models	[112]
Evf2	HD and epilepsy	*Dlx5/6*	Rabbit models and cell lines (mouse EL250, PL253 and PL452)	[120]
utNgn1	Neurogenesis	*Neurog1*	Mice models and ESC lines	[126]
IRENEs	CHD	*Nkx2-5*	Mice models and cell lines (human RUES2 cells and iPSC-derived CMs, mouse HL-1)	[137]
Wisper	AOS and CM	*Wisp2*	Patient samples and mice models	[140]
HERNA1	AS and HCM	*synaptotagmin XVII* and *SMG1*	Patient samples, mice models and cell lines (HEK-293T cells)	[142]
Bvht	CVD	*Mesp1*	ESC lines	[144]
Meteor	CVD	*Eomes*, *T*, *Gsc*, *Gata4* and *Isl1*	Mice models and ESC lines	[145]
Lnc-leptin	Obesity	*Lep*	Mice models	[149]
OLMALINC	Hepatic steatosis and NAFLD	*SCD*	Patient samples and cell lines (HepG2 and Fa2N4)	[153]
LncASIR	T2D	*PI3K*, *Fabp4*, *Glut4* and *Srebp1c*	Mice models	[164]

## Data Availability

Not applicable.

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
