# Peer review of "Enhancer RNA (eRNA) in Human Diseases"

_ijms, 2022, doi:10.3390/ijms231911582_

Round 1

Reviewer 1 Report

The authors present a review on "Enhancer RNA (eRNA) in Human Diseases" and attempt to bring a holistic view of enhancers in various facets of diseases, viz. metabolic, cancers, neurodenegenrative, cardiovascular diseases.   The review is succinctky mentioned supplemented with figures and texts.
The conclusions sets a good standard describing the role of the eenhancers in the aforementioned diseases. 
While the authors attempt to narrate the regulatory nature of enhancers and upstream regions of the gene that the enhancers are associated with, there must be a distinct epithet or unforimity for all diseases.  A more detailed structured format for enhancers, their effective role on CTCFs, other regulons/regulatory elements would be an asset for thsi review. This paragraph will accomplish reading style and bridge the gap for other diseases. 
  Pl find attached subtle edits
Over the years, substantial enhancers have been categorically studied in rare diseases and compelx diseases includingmetabolic diseases such as Lipodystrophy which the authors can add a couple of paragraphs in these directions. 
The figure 1 is very nice, but an overview of Enhancers and their role could be depicted
There must be a table as well showcasing the regulatory nature of all Enhancers

Scores on a scale of 0-5 with 5 being the best 

Language: 3.5
Novelty: 3
Scope and relevance: 3
Brevity:3

Author Response

We thank the overall positive comments from Reviewer #1. Please see attachment for our point to point response. 

Reviewer 2 Report

Enhancer RNAs (eRNAs) form a class of non-coding RNAs with roles in disease.  This review summarizes some aspects of eRNAs with disease including cancer. Overall,  the manuscript covers several important disease areas and presents a clear picture of eRNA involvement.

The manuscript has many scientifically inappropriate descriptions throughout. For example,  in the abstract 'wildly', 'shreds', 'treating diseases'; all are not appropriate. There are statements lack context and substance, i.e., treating disease with eRNAs? or an assortment of human disease, but really a few examples of several different cell based disease models. Also, I don't see how this manuscript adds much beyond several recently  published reviews.

The  distinction between open chromatin, global gene expression enhancer regulation (disease-specific) and other aspects of collective eRNA biology is lacking. The manuscript rather just highlights some examples for different disease but fails to integrate important molecular concepts. This  is particularly true for the GRN introduction and description. 

Line 66-67; rewrite no colon

Line 69  'golden'? 

Line 74; Figure 3 referenced before Figure 1

Author Response

We thank comments from reviewer #2. Please see attachment for our response. 

Round 2

Reviewer 2 Report

I am in favor of accepting the manuscript with minor English modifications.

It is recommended to find a native English speaker to check the full text.

Author Response

Response: We thank the reviewer’s valuable comments. In this version, we revised our manuscript very carefully to addressed typos. In addition, we asked Dr. Libing Shen for English editing throughout our manuscript. Please see below for some examples.

Line 471: from “Investigating and exerting eRNAs in metabolic diseases, particularly T2D, appears to be in various ways” to “Investigating the eRNAs in metabolic diseases, particularly T2D, shows that they are involved in various metabolic pathways”.

And line 515: from “That draws a type of serial influence that eRNAs may trigger in different pathologies and demonstrates the critical involvement eRNAs have in complex biological processes” to “That draws a series of attention that eRNAs are involved in multiple biological processes and demonstrates their critical roles in different human diseases”.

Please see all other modifications in our revised manuscript.
